POINT OF VIEW

# Biological causal links on physiological and evolutionary time scales

**Abstract** Correlation does not imply causation. If two variables, say A and B, are correlated, it could be because A causes B, or that B causes A, or because a third factor affects them both. We suggest that in many cases in biology, the causal link might be bi-directional: A causes B through a fast-acting physiological process, while B causes A through a slowly accumulating evolutionary process. Furthermore, many trained biologists tend to consistently focus at first on the fast-acting direction, and overlook the slower process in the opposite direction. We analyse several examples from modern biology that demonstrate this bias (codon usage optimality and gene expression, gene duplication and genetic dispensability, stem cell division and cancer risk, and the microbiome and host metabolism) and also discuss an example from linguistics. These examples demonstrate mutual effects between the fast physiological processes and the slow evolutionary ones. We believe that building awareness of inference biases among biologists who tend to prefer one causal direction over another could improve scientific reasoning.

**AMIT KARMON AND YITZHAK PILPEL**

More than 50 years ago Ernst Mayr introduced the distinction between "proximate" and "ultimate" causes in biology (*Mayr, 1961*). In this now classical theory, proximate causes act via short-term physiological processes, and ultimate causes act via long-term evolutionary processes. Put simply, the proximate cause addresses the question of "how?" and the ultimate cause addresses the "why?".

The extensive data yielded by modern biology are full of intriguing correlations between variables. A great conceptual challenge, which often triggers intense arguments, is to decipher the nature of causal links that underlie these observations. It seems to us that biologists tend to interpret causation in a way that reflects their education and other biases, often noticing at first the proximate cause. We follow with a few examples of biological questions in which the direction of a causal link (physiological or evolutionary) has been, or could have been, debated. In each example we describe how a physiological causal link ("A causes B") could account for

an observed correlation, but also how the opposite causal direction ("B causes A"), which takes place over an evolutionary time scale, can and perhaps should be invoked. We conclude with a discussion on the interaction between the fast-acting and slow-acting effects in biology and ask why biologists might intuitively focus on one effect over another.

## Codon usage optimality and gene expression

Let's first consider the often-observed correlation between the codon usage of a gene and its protein expression level (*Sharp and Li, 1987*). Scientists who wish to overexpress a gene in a heterologous system know that they need to "codon-optimize" their gene. That is to say, they must alter its coding sequence so that it is the best match for the codon usage of the most highly expressed genes in the host genome. These are often the codons that correspond to abundant tRNAs in that species. Indeed the effects of codon optimization of gene expression are most pronounced in heterologous expression

Many causal links in biology might be caused by fast-acting physiological processes acting in one direction and slower evolutionary processes acting in the opposite direction.
Illustration: Claudia Stocker, vividbiology.com

systems (see *Gustafsson et al., 2004*). Yet also for the natural genes in a genome, the most codon-optimized genes are often most highly used and expressed (*Sharp and Li, 1987*; *Man and Pilpel, 2007*). This may lead to the conclusion that optimal codon usage is a cause of high protein expression for natural genes too. In other words, codon optimization is a proximate cause for protein expression level.

But could the opposite causal direction provide additional explanation of the observed correlation? Let's state clearly the potential opposite effect: "a high level of expression of a protein (that may have been achieved via various mechanisms) affects the codon optimality of its gene". Is that possible? On an evolutionary time scale, maybe it is and protein expression level is an ultimate cause of codon optimality over millions of years (*Kudla et al., 2009*). It is plausible that highly expressed genes are selected more vigorously for codon optimization than lowly expressed genes, because the fitness cost of not optimizing them (in terms of a burden on the translation apparatus) is likely to be higher. We can even imagine that highly expressed genes could affect the translation machinery (e.g. the tRNA pool) to become more optimally suited to their codons, and thus force the genome to

define their codon usage as "optimal" (*Yona et al., 2013*).

While the evolutionary causal direction in this example was well appreciated early on mainly by evolutionary biologists (*Ikemura, 1981*; see *Plotkin et al. 2011* for a summary), for others the physiological causal direction was the first to be envisaged (*Sharp et al., 1986*).

## Gene duplication and genetic dispensability

Our second example has to do with gene duplication, and the essentiality of the same genes. Gene duplication has without doubt played a major role in the evolution of biological complexity and has served as a source of genetic novelty (*Ohno et al., 1968*; *Lynch and Conery, 2000*). Notwithstanding, a well-known observation is that genes that have "duplicates" (or paralogs) in the genome often appear to be less essential, compared to genes that occur as a single copy (*Gu et al., 2003*). An intuitive interpretation of this correlation, that was shown to be correct in several cases, suggests that a gene is non-essential because its duplicate can compensate if it is mutated (*Kafri et al., 2005*, *DeLuna et al., 2010*). In other words, duplication causes dispensability. But could it be that dispensability causes duplication too? How could that happen?

Consider the possibility that "dispensable" genes, defined as those that can be deleted without reducing the fitness of the organism, may also be tolerated at a duplicated dosage. On the other hand, "essential" genes, whose copy number cannot be reduced to zero, may also not tolerated at more than one copy (*He and Zhang, 2006*). According to this possibility, if a gene is dispensable, the chances that evolution will tolerate its duplication are higher compared to genes that carry out more essential functions. Please note that we do not intend to judge the plausibility of this alternative evolutionary explanation here (see Discussion below), but instead we want to show that this alternate explanation is certainly of interest and yet it took longer to appear.

What is common to the codon usage and gene essentiality examples above? Both feature a strong correlation observed throughout genomes. In both cases, the originally proposed effect, is perhaps more intuitive to many molecular biologists and describes a physiological process that typically occurs on a time scale of minutes, hours or, at the most, days (i.e., protein expression and compensation between genes). In contrast, the alternatives – "high expression causes codon optimality" and "gene dispensability causes duplication" – both describe processes that could occur on a time scale of millions of years or more.

Let's now take two additional examples from biology, and then one from linguistics.

## Stem cell division and cancer risk

Recently, the number of stem cell divisions within a tissue or organ in humans was shown to correlate strongly with the lifetime risk of cancer arising within that body part (*Tomasetti and Vogelstein, 2015*). The interpretation given to this finding was that each stem cell division has a certain probability to result in a carcinogenic mutation such that tissues that divide more have a greater chance of developing these mutations. Essentially this interpretation proposes that stem cell division affects risk of cancer. But can we also consider the flip side of this effect?

In a tissue with a cell count that remains steady, a high rate of stem cell divisions should be balanced with a high rate of cell death. Suppose also that certain tissues are more prone than others to becoming cancerous simply because, for example, they are commonly exposed to environmental damage that could mutate the DNA in the non-dividing cells. We

would expect such tissues, which would include the skin and those in the gut, to have evolved mechanisms to defend themselves against cancer. Maybe instead of stem cell division affecting risk of cancer we are seeing the results of such a protective mechanism. That is to say, high-risk tissues feature high rates of cellular turnover, with extensive programmed cell death to eliminate potentially mutated cells and many stem cell divisions to compensate for the loss of cells.

Interestingly, in this example, the physiological causal link again appears to have been more intuitive for many biologists, and the evolutionary one is provided here as an additional possibility for further discussion. Of course, this proposed mechanism might have a cost too, because more stem cell divisions means more DNA replication and an increased risk of cancerous mutations. Nevertheless, perhaps this cost is worth paying to eliminate the cells damaged by the environment.

## The microbiome and host metabolism

Ground-breaking experiments on microbiome manipulation in mice showed that colonization of germ-free mice with a microbiota of obese donors results, within a couple of weeks, in a greater increase in total body fat compared to colonization with a 'lean microbiota' (*Turnbaugh et al., 2006*). These experiments showed that the microbiota can affect the host metabolic environment, perhaps for their own benefit (see also *McNally and Brown, 2015*). Put in simpler terms, the microbiota composition might cause host obesity, potentially serving the microbes' needs. Importantly, the seminal work by Turnbaugh et al. went well beyond merely observing a correlation; it also established a direct causal link through experimental manipulation.

Notwithstanding the validity of this study, and in no way implying that the original causal link is wrong, what if we were to also consider a slow-acting effect in the opposite direction? Could a host's inner metabolic environment select for certain microbes? This possibility opens up others. "Obese" or "lean" microbiota might thrive in "obese" or "lean" hosts, and thus they might have evolved adaptive mechanisms to help design a comfortable environment for themselves. In addition, other factors in a person's metabolism might determine which germs will eventually colonize them. In other words, perhaps your body selects its microbiota,

which in turn feedbacks onto your metabolism. This would be a slow evolutionary effect, at least in terms of microbial evolutionary time scales.

What experiments would support this conjecture? Maybe the above-mentioned experiment of microbiota transfer might provide an answer if it was extended over months or years. For example, take an originally lean mouse, make a germ-free version of it, and transplant into it the microbiota of either a lean or an obese mouse donor. Then, follow the stability of the transplants for prolonged periods of time. If metabolic environment affects the microbiome, then this originally lean mouse should more stably sustain the donation from the lean mouse. Interestingly, this alternative causal direction is supported by a recent study that compared the microbiome of hundreds of human twins and concluded "host genetics influence the composition of the human gut microbiome and can do so in ways that impact host metabolism" (*Goodrich et al., 2014*). Again in this example, it is important to note that the originally proposed effect occurs within a short-term physiological time frame, yet a plausible effect in the opposite direction might play out over a microbial evolutionary time scale.

## Language and perception

Our final example is one from language and linguistics. The principle of "linguistic relativity" holds that the structure of a language affects the ways in which its speakers conceptualize their world, or, put more simply: "we think in the way that we speak". There is circumstantial evidence that serves to provide support for this notion. For example, native speakers of the Pirahã language, which does not have names for numbers above two, perform relatively poorly in simple counting tasks (*Everett, 2005*); natives of languages in which directions are absolute ("south", "north," "east" and "west") rather than relative ("right" or "left") navigate better (*Boroditsky, 2011*); and natives of languages such as Russian, in which different shades of blue have different names, can distinguish between such colours more easily (*Winawer et al., 2007*). These and other examples indeed suggest that the way people speak affects the way they think, or perceive the world. Yet here too, we could consider an effect in the opposite direction. Put explicitly: maybe the way we think or perceive the world affects the way our language evolves. The linguist John McWhorter also strongly

argues against "linguistic relativity," suggesting that, despite differences between languages, "the world looks the same in any language.

Like in the above examples from biology, there are two potential directions of effect that could explain correlations between language and perception, and they would act on either a short time scale or a long time scale. The putative effect of perception on language would necessarily take place over an evolutionary time scale, while effects of language on perception would occur during our life span as we acquire a language. For example, is it possible that the language of the Aboriginal group Kuuk Thaayorre, which uses absolute directions, was shaped in its rather peculiar way because absolute directions were more evolutionarily critical than relative directions in that culture? That is, maybe this group, to which navigation through absolute directions comes naturally, survived better over the years, and their language simply reflects their natural inclinations. Likewise, the different names for shades of blue in Russian might reflect a peculiarity in the genes that determine colour vision in the Russian population. Although we will leave the resolution of this general dispute to the linguists, it is interesting to notice that, in this example too, the observed correlation between language and perception might be interpreted as indicating a bi-directional causal link.

## Discussion

The five examples described above each involve a pair of variables which are correlated. In each case, a short-term effect that acts on a physiological time scale seemed more intuitive at first, at least for traditionally trained molecular biologists. Curiously, the slower-acting potential effect that requires involvement of evolutionary processes was less intuitive, albeit typically not to evolutionary biologists. We note though that under certain conditions a bi-directional effect, both physiological and evolutionary might be at work. In particular, consider again A and B as two characteristics of an organism. First, suppose that B is a characteristic that increases an organism's fitness (e.g. high expression of a gene). Second, suppose that the organism can display B without the characteristic A (e.g. high codon optimality of that gene), but that A makes B more efficient. Then, after a long evolutionary process, we would expect the organism to develop a mechanism in which B is established through the means of A. This means that the

effect "B causes A" depends on the existing effect "A causes B". For example, evolution likely selected for high codon optimality of highly expressed genes (*Kudla et al., 2009*). But selection for optimal codons could only come into play more effectively in highly expressed genes if codon optimization affected gene expression in the first place.

Let us return to Mayr's classic that introduced the distinction between proximate and ultimate causal explanations (*Mayr, 1961*). This distinction has been debated over the years. Some have argued later that the distinction is not always a sharp one, and that "proximate mechanisms both shape and respond to selection" (*Laland et al., 2011*). Without committing ourselves to any claim about the validity, sharpness or universality of this distinction, we agree that sometimes two variables can be the proximate and the ultimate causes of one another. While A is a proximate cause of B, B may have prevailed even before A, and may have ultimately affected A.

So why is the reasoning of many biologists seemingly more prone to focus at first on the effect acting on the short-term, physiological time scale explanation and not on the processes that take millions of years to manifest themselves? Is it because of the biologists' training? Or is it because it's easier to think in terms of attainable laboratory experiments? Other reasoning concepts and tools of course must be exercised too. For example, Occam's Razor – the problem solving principle in philosophy that suggests that the theory with fewest assumptions should be preferred at first – could be helpful here in judging each potential causal direction. For example in the gene duplication and genetic dispensability problem presented above, the evolutionary causal effect requires the extra assumption that dispensable genes are more likely to undergo gene duplication during evolution; this is non-trivial.

In any case, it is interesting to realize that our reasoning is influenced by certain biases that might prevent us, in some cases, from seeing the full picture. We hope that an awareness of such pitfalls could help others to fix or even avoid them.

## Acknowledgements

The authors thank the Minerva Center for Live Emulation of Evolution in the Lab for grant support.

Amit Karmon is in the Department of Molecular Genetics, Weizmann Institute of Science, Rehovot, Israel

http://orcid.org/0000-0003-0763-6696

Yitzhak Pilpel is in the Department of Molecular Genetics, Weizmann Institute of Science, Rehovot, Israel

Pilpel@weizmann.ac.il

http://orcid.org/0000-0003-3200-9344

**Competing interests:** The authors declare that no competing interests exist.

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
