## [Decision Letter]

Thank you for submitting your article "Biological causal links on physiological and evolutionary time scales" to *eLife* for consideration as a Feature Article. Two peer reviewers have reviewed your article, and an Associate Features Editor – Stuart King – and the Features Editor – Peter Rodgers – have overseen the evaluation.

The Associate Features Editor has drafted this consolidated decision letter to help you prepare a revised submission.

Summary:

Karmon and Pilpel present a highly unusual opinion piece that reads more like a scientist's musings or even a stream of consciousness, rather than as a comprehensive review of a topic. However, as unusual as this contribution is, it does make a noteworthy and thought-provoking point: that evolutionary explanations of biological causality are often overlooked by researchers focusing on fast-acting physiological explanations.

The five sub-sections selected by Karmon and Pilpel are also varied: some cover well-trodden ground (but are nonetheless worth repeating), while others seem almost entirely new. For example, the sub-section about stem cell division and cancer risk seems to be a mostly new idea that has not been discussed by evolutionary biologists so much or as long as codon usage and gene duplication; it thus provides good contrast in the article.

Together it is an interesting read, but lacks scholarship and connection to the prior literature. The following issues need to be remedied in the revision.

1) Connection to the prior literature

1A) More than fifty years ago, Ernst Mayr wrote about the nature of causation in biology, and distinguished between proximate (physiology) and ultimate causes (evolution). Since then, others have noted that this rigorous distinction is problematic to say the least, and that reciprocal causation may now prove useful for many specific biological problems (Laland et al. 2011; DOI: 10.1126/science.1210879). Since your paper attempts to discuss well-defined segments of this problem, you should put your arguments and speculation within the proper historical context.

1B) A few other important citations are missing from specific sub-sections. In the gene duplication example, a reference to Ohno (e.g. S. Ohno, Evolution by Gene Duplication, Springer 1970) is critical but missing, and a reference to Lynch's work on gene duplication and genome evolution (e.g. M. Lynch, The Origins of Genome Architecture, Sinauer 2007) should be added too. In the microbiome example, a reference to the large evolutionary literature on "niche construction" would be useful.

2) Consideration of problem-solving principles

At the end of the paper, you ask "why is the reasoning of many biologists, so it seems, more prone to prefer the effect acting on the short-term, physiological time scale explanation over the processes that take millions of years to manifest themselves?" The brief discussion that follows could be improved if it also considered how general problem-solving principles, such as Occam's razor, could explain the 'bias'.

For example, in the gene duplicability sub-section – the pattern that needs to be explained is that duplicated genes in the genome often appear to be less essential. There are two, mutually non-exclusive explanations:

i) The presence of a back-up gene mitigates fitness effect of mutations in the other copy.ii) Gene dispensability drives gene duplication on an evolutionary timescale.

Both hypotheses are relevant, but there is a good reason why researchers should start by focusing on the first hypothesis. This is because the second hypothesis has an extra assumption (that dispensable genes are more likely to undergo gene duplication during evolution), which is non-trivial and has relatively little empirical basis. The first hypothesis has no such hidden assumption, and testing its predictions is straightforward (e.g. by comparing the fitness of singleton and double-knock-outs). Therefore, only if the predictions of this first hypothesis fail would most people consider the more complicated models that assume that evolution (i.e. gene duplicability) differs across gene classes. By not taking this into consideration, there is a risk that the current manuscript could potentially encourage readers to ignore simple null models.

3) Example of codon usage

The example about codon usage is perhaps the best illustration of the overall point made in the piece. However, the article should make it clearer that the fast-acting physiological effect is generally only visible in the unnatural case of a highly expressed transgene. By contrast, the effects of codon usage on synthesis for endogenous genes are much weaker, even for the most strongly expressed endogenous genes, which is why it has taken evolutionary timescales for the genome to adjust codon usage (and has done so only for high-expression endogenous genes).

---

## [Author Response]

*The following issues need to be remedied in the revision. 1) Connection to the prior literature 1A) More than fifty years ago, Ernst Mayr wrote about the nature of causation in biology, and distinguished between proximate (physiology) and ultimate causes (evolution). Since then, others have noted that this rigorous distinction is problematic to say the least, and that reciprocal causation may now prove useful for many specific biological problems (Laland et al. 2011; DOI: 10.1126/science.1210879). Since your paper attempts to discuss well-defined segments of this problem, you should put your arguments and speculation within the proper historical context.*

Thank you for this essential point. We now made explicit reference to Mayr’s classic, once at the Introduction, it now opens our manuscript by introducing the terms proximate and ultimate, and then again in the Discussion, where we also discuss Laland’s reflection. See “More than 50 years ago Ernst Mayr…” and “Let us return back to Mayr’s classic”.

*1B) A few other important citations are missing from specific sub-sections. In the gene duplication example, a reference to Ohno (e.g. S. Ohno, Evolution by Gene Duplication, Springer 1970) is critical but missing, and a reference to Lynch's work on gene duplication and genome evolution (e.g. M. Lynch, The Origins of Genome Architecture, Sinauer 2007) should be added too. In the microbiome example, a reference to the large evolutionary literature on "niche construction" would be useful.*

Correct. We now reference Ohno and Lynch in the gene duplication section, and provide a representative recent mention of niche construction, which we connect to our argument (see “Gene duplication, no doubt, has played…” and These experiments showed that the microbiota can affect the host metabolic environment, perhaps for their own benefit (see also [15])). We also reference codon optimization in the context of heterologous gene expression (see point 3) below).

*2) Consideration of problem-solving principles At the end of the paper, you ask "why is the reasoning of many biologists, so it seems, more prone to prefer the effect acting on the short-term, physiological time scale explanation over the processes that take millions of years to manifest themselves?" The brief discussion that follows could be improved if it also considered how general problem-solving principles, such as Occam's razor, could explain the 'bias'. For example, in the gene duplicability sub-section – the pattern that needs to be explained is that duplicated genes in the genome often appear to be less essential. There are two, mutually non-exclusive explanations:*

*i) The presence of a back-up gene mitigates fitness effect of mutations in the other copy.ii) Gene dispensability drives gene duplication on an evolutionary timescale.*

*Both hypotheses are relevant, but there is a good reason why researchers should start by focusing on the first hypothesis. This is because the second hypothesis has an extra assumption (that dispensable genes are more likely to undergo gene duplication during evolution), which is non-trivial and has relatively little empirical basis. The first hypothesis has no such hidden assumption, and testing its predictions is straightforward (e.g. by comparing the fitness of singleton and double-knock-outs). Therefore, only if the predictions of this first hypothesis fail would most people consider the more complicated models that assume that evolution (i.e. gene duplicability) differs across gene classes. By not taking this into consideration, there is a risk that the current manuscript could potentially encourage readers to ignore simple null models.*

Thank you for a very good point. We now refer to Occam’s Razor as an additional concept and tool in prioritizing between theories (causal effects, in our case) and indeed follow this exact argument to show how it could be used to support one causal direction over then other (see “Other reasoning concepts and tools of course must be exercised too. For example, Occam’s Razor…”). I believe that with that said the reader might easily apply these reasoning tools to additional problems.

3) Example of codon usage The example about codon usage is perhaps the best illustration of the overall point made in the piece. However, the article should make it clearer that the fast-acting physiological effect is generally only visible in the unnatural case of a highly expressed transgene. By contrast, the effects of codon usage on synthesis for endogenous genes are much weaker, even for the most strongly expressed endogenous genes, which is why it has taken evolutionary timescales for the genome to adjust codon usage (and has done so only for high-expression endogenous genes).

We agree with the referee and indeed now clarify that the physiological direction applies mainly for heterologous expression systems (“Indeed the effects of codon optimization of gene expression are most pronounced in heterologous…”).